# The Long-Term Persistence of *Borrelia burgdorferi* Antigens and DNA in the Tissues of a Patient with Lyme Disease

**DOI:** 10.3390/antibiotics8040183

**Published:** 2019-10-11

**Authors:** Eva Sapi, Rumanah S. Kasliwala, Hebo Ismail, Jason P. Torres, Michael Oldakowski, Sarah Markland, Gauri Gaur, Anthony Melillo, Klaus Eisendle, Kenneth B. Liegner, Jenny Libien, James E. Goldman

**Affiliations:** 1Department of Biology and Environmental Science, University of New Haven, West Haven, CT 06516, USAheboismail@yahoo.com (H.I.);; 2Central Teaching Hospital Bolzano L Böhlerstr, 539100 Bolzano, Italy; 3Private practice, 592 Route 22, Suite 1B, Pawling, NY 12564, USA; 4Northwell System, Northern Westchester Hospital, Mount Kisco, NY 10549, USA; 5Health Quest System, Sharon Hospital, Sharon, CT 06069, USA; 6Department of Pathology, SUNY Downstate Health Sciences University, Brooklyn, NY 11203, USA; 7Department of Pathology and Cell Biology, Columbia University, New York, NY 10031, USA; jeg5@cumc.columbia.edu

**Keywords:** *Borrelia burgdorferi*, spirochete, Lyme disease, persisters, biofilms, antibiotic resistance

## Abstract

Whether *Borrelia burgdorferi,* the causative agent of Lyme disease, can persist for long periods in the human body has been a controversial question. The objective of this study was to see if we could find *B. burgdorferi* in a Lyme disease patient after a long clinical course and after long-term antibiotic treatment. Therefore, we investigated the potential presence of *B. burgdorferi* antigens and DNA in human autopsy tissues from a well-documented serum-, PCR-, and culture-positive Lyme disease patient, a 53-year-old female from northern Westchester County in the lower Hudson Valley Region of New York State, who had received extensive antibiotic treatments during extensive antibiotic treatments over the course of her 16-year-long illness. We also asked what form the organism might take, with special interest in the recently found antibiotic-resistant aggregate form, biofilm. We also examined the host tissues for the presence of inflammatory markers such as CD3+ T lymphocytes. Autopsy tissue sections of the brain, heart, kidney, and liver were analyzed by histological and immunohistochemical methods (IHC), confocal microscopy, fluorescent in situ hybridization (FISH), polymerase chain reaction (PCR), and whole-genome sequencing (WGS)/metagenomics. We found significant pathological changes, including borrelial spirochetal clusters, in all of the organs using IHC combined with confocal microscopy. The aggregates contained a well-established biofilm marker, alginate, on their surfaces, suggesting they are true biofilm. We found *B. burgdorferi* DNA by FISH, polymerase chain reaction (PCR), and an independent verification by WGS/metagenomics, which resulted in the detection of *B. burgdorferi* sensu stricto specific DNA sequences. IHC analyses showed significant numbers of infiltrating CD3+ T lymphocytes present next to *B. burgdorferi* biofilms. In summary, we provide several lines of evidence that suggest that *B. burgdorferi* can persist in the human body, not only in the spirochetal but also in the antibiotic-resistant biofilm form, even after long-term antibiotic treatment. The presence of infiltrating lymphocytes in the vicinity of *B. burgdorferi* biofilms suggests that the organism in biofilm form might trigger chronic inflammation.

## 1. Introduction

Lyme borreliosis, a tick-borne multisystemic illness caused by the spirochete *Borrelia burgdorferi* sensu lato, has grown into a major public health problem [1,2,3,4,5]. The frontline treatment for Lyme disease is the administration of antibiotics [6,7,8]. However, relapse of the disease often occurs, with lingering symptoms when treatment is discontinued, regardless of the choice of antibiotics [9,10,11,12,13,14]. The question is whether or not the persistent symptoms are related to ongoing spirochetal infection despite antibiotic therapy [15]. In vivo animal studies with immunocompetent mice, dog, and non-human primate models have identified a surviving, but not cultivatable form of *B. burgdorferi* that can withstand antibiotic treatment [16,17,18,19,20]. Recent studies on rhesus macaques and mice confirmed these observations by demonstrating a metabolically active, persistent *B. burgdorferi* post-treatment in antibiotic-treated animals [20,21,22,23,24]. There are also human studies providing clinical evidence that a chronic form of Lyme disease could be caused by a persistent spirochetal infection, which could explain the lingering symptoms [25,26,27,28].

It has been previously suggested that antibiotic-resistant forms of *Borrelia* might be due to the formation of alternative morphologies, such as “round bodies” (cyst and granules), which were first identified and studied in the 1990s by Drs. Øystein Brorson and Alan MacDonald [29,30]. Effective antimicrobial agents for the round bodies [31], however, did not produce better in vivo outcomes, suggesting that there might be additional alternative forms providing refuge from antibiotics [32].

One such alternative form is called biofilm. Bacterial biofilms are responsible for several chronic diseases such as periodontitis, osteomyelitis, bacterial endocarditis, and even chronic lung infection in cystic fibrosis patients [33,34,35,36,37]. Organisms in biofilms are particularly difficult to eliminate because their antibiotic resistance can be as much as 1000-fold greater than that of their free-living counterparts [35,36,37]. The biofilm resistance is based upon multiple mechanisms, such as incomplete penetration of the antibiotics into the matrix, inactivation of antibiotics by altered microenvironment within the biofilm, and a highly protected, resistant bacterial population [38,39,40]. In the past few years we have provided substantial evidence that *B. burgdorferi* sensu lato and stricto are indeed capable of forming biofilms in vivo and in vitro [41,42,43]. Many of the important biofilm traits like structural rearrangements in the biofilm and the development of protective matrices on the surface were identified not just in cultured spirochetes but also in human skin borrelial lymphocytoma biopsy tissues [41,42,43]. Specific biofilm markers, such as characteristic protective layers containing alginate, extracellular DNA, and calcium channels and protrusions, can be found in *Borrelia* biofilm in vitro and in vivo [41,42,43]. We have also reported that B. burgdorferi biofilm formation dramatically increases the resistance to antibiotics [44,45,46]. Taken together, these observations strongly suggest that biofilm formation by *B. burgdorferi* could play a significant role in their survival in diverse environmental conditions by providing refuge for individual cells. The formation of biofilm by *B. burgdorferi* could also change the way we think about Lyme disease, especially in patients where infection seems to persist despite long-term antibiotic treatment.

Here we report a patient with serum-, PCR-, and culture-positive Lyme disease who had received extensive antibiotic treatment over 16 years. We asked if we could find evidence of *Borrelia* after the antibiotic therapies and, if so, what form the organism might take. We explored the hypothesis that antibiotic-resistant forms of *B. burgdorferi* might exist in *Borrelia*-infected human autopsy tissues and that the persistent infection might be associated with the corresponding pathological findings. Therefore, we evaluated autopsy tissues from this patient for pathological findings and for the presence of borrelial spirochetes and biofilms using immunohistochemical methods (IHC), fluorescent in situ hybridization (FISH), polymerase chain reaction (PCR), and whole-genome sequencing (WGS)/metagenomics methods. Additionally, the presence of inflammatory infiltrates containing CD3+ T cells and their relationship with *Borrelia*-positive cells and aggregates was also studied.

### 2.1. Clinical History

A 39-year-old woman with at least a two-year history of spastic paraparesis, cranial nerve palsies, and lymphocytic CSF pleocytosis was first evaluated by one of the authors (K.L.) in 1989. The patient had lived in a tick-infested cottage community in northern Westchester County in the lower Hudson Valley Region of New York State, a highly epidemic area for Lyme disease. She also resided in northern California. There was no history of tick attachment or of eruptions suspicious for erythema migrans. A thorough evaluation for Lyme disease was entirely negative. Splenectomy had been performed for idiopathic thrombocytopenic purpura in 1976.

She was treated empirically for Lyme disease with 21 days of intravenous cefotaxime (2 g, every 8 h) in April 1990. CSF pleocytosis prior to and following treatment was unchanged, as was her clinical status. She did not improve after four months of oral minocycline. She became home- and wheelchair-bound. She did not have any pets. It is unlikely that she was infected in the interim between April 1990 and December 1991. CSF was re-examined in December 1991 and revealed a lymphocytic pleocytosis. Spinal fluid placed in BSK-II medium and cultured at the Centers for Disease Control, (Fort Collins, CO, USA) yielded a growth of spirochetes several weeks later, confirmed by PCR to be *B. burgdorferi*. A “pulse” regimen of cefotaxime starting in January 1992 (4 g, every 8 × 3 consecutive doses, once weekly for 13 weeks) resolved the pleocytosis. Her clinical status and gait improved modestly. The “pulse” regimen was intensified (cefotaxime 4 g every 8 h × 6 consecutive doses, weekly) for 10 months, but her condition slowly worsened. Intravenous antibiotic therapy was discontinued and she was transferred to the Mayo Clinic, (Rochester, MN, USA) for consideration of the placement of a baclofen pump in view of her lower extremity spasticity and for a further opinion regarding management. A test dose of baclofen, although reducing spasticity, made it impossible for her to stand or transfer, so this approach was abandoned. In January 1993, she was treated with 10 days of intravenous ceftriaxone and, in view of a positive ANA at high titer (1:2560), with intravenous methylprednisolone followed by oral prednisone for six weeks.

In May 1993, she became unable to hold a cup, roll over in bed, or remember conversations held minutes earlier. A CT scan of the chest revealed pleuropericardial effusions. A pericardial window was created. A touch-preparation of the pericardium revealed a *Borrelia*-compatible spirochete and histology showed pericarditis with infiltration by macrophages, lymphocytes, and plasma cells. A phycoerythrin stain showed spirochetal-compatible structures.

Treatment with 109 continuous days of intravenous cefotaxime (2 g, IV every 8 h) resulted in dramatic improvement in encephalopathy and enabled her to ambulate 500 feet with a rolling walker. The pleuropericardial effusions resolved. Daily intravenous cefotaxime was continued for an additional three months. In the ensuing months off treatment, she became increasingly encephalopathic. Daily intravenous cefotaxime was re-instituted in October 1994. 

Neuropsychological testing undertaken before and after four months of this treatment demonstrated a modest objective improvement in cognitive function. This patient was a well-functioning and highly valued pediatric intensive care nurse prior to falling ill. She underwent formal detailed neuropsychological testing in October 1994 and again in February 1995 in order to assess her cognitive status and the impact, if any, of treatment with a four-month course of intravenous cefotaxime (2 g every 8 h). On objective neuropsychological tests, she showed marked memory deficits. Following treatment, she showed a modest improvement in visual memory, delayed recall of verbal memory, and meaningfully increased scores in attention and concentration. Despite the applied treatment, she continued to have profound deficits in memory, in striking contrast to her overall verbal intelligence, which, on testing, fell within the superior range.

Changes in the patient’s insurer’s policies for reimbursement prevented further antibiotic treatment and her condition deteriorated. Serial CSF examinations revealed variable pleocytosis. Lyme ELISA, which had seroconverted in December 1991, subsequently was serovariable, and Western blots, although showing bands of high specificity for Lyme disease, often failed to satisfy the CDC criteria for five out of 10 CDC-specific IgG bands. *B. burgdorferi* DNA was demonstrated by PCR in the patient’s blood on multiple occasions between 1995 and 2001. 

Treatment with intravenous ceftriaxone (4 g, three days per week), donated by the manufacturer, in late September 2000 and mid-March 2001, with some hiatuses, for a total of 36 administered doses, did not yield an apparent clinical benefit.

Following a change in her insurance to public health benefits, intravenous cefotaxime 2 grams three times per day was applied for three weeks starting on 27 May 2003, but was suspended because there was no mechanism for reimbursement to the facility. While attempts to secure means of reimbursement were underway, on 6 July 2003, and despite therapeutic diphenylhydantoin levels for a seizure disorder that had developed earlier in the course of her illness (onset: May 2000), she sustained a series of grand mal seizures and was transported by ambulance to a local hospital. Seizures were controlled with the adjustment of anti-epileptic therapy, but hypotension developed. After discussion between the hospital physicians and the next of kin, a Do Not Resuscitate Order was issued. Shortly thereafter, the patient expired.

The patient’s case through 1996 has been previously reported [11,47,48]. Detailed documentation of her case, including correspondence concerning her care and the circumstances leading up to her death, has been published [49].

## 3. Results

### 3.1. Pathological Findings—CNS and PNS

The brain leptomeninges contained infiltrates composed of chronic inflammatory cells, principally lymphocytes, plasma cells, and macrophages (Figure 1a). A number of meningeal vessels had been occluded and recanalized (Figure 1b), but there were no acutely occluded vessels. Blood vessels in the cortex, subcortical white matter, basal ganglia, and thalamus displayed perivascular accumulations of lymphocytes and macrophages, with rare plasma cells (Figure 1d), but no evidence of an acute vasculitis. The parenchyma contained a large number of small infarcts, affecting both gray and white matter of the hemispheres (Figure 1c,e and Figure 2a,b). Although we found these pathological features in many areas of the hemispheres, the frontal cortex and subcortical white matter were the most severely affected. Infarcts were present in the internal capsule, which produced a degeneration of the corticospinal tract, visualized with a Bielschowsky silver stain of the spinal cord (Figure 2d). Infarcts were also present in the cerebellum (Figure 2c). We found chronic inflammatory infiltrates around blood vessels in cranial nerves (Figure 2e). There was no indication of cranial nerve degeneration, based on myelin or silver stains in our samples. 

### 3.2. Pathological Findings—Kidney/Heart/Liver

The kidney contained many foci of lymphocyte and plasma cell infiltrates (Figure 3), involving both the cortex and the medulla. Scattered blood vessels were chronically occluded, and glomeruli were sclerotic (Figure 4). In addition, there were fibrous scars.

Heart sections of left and right ventricles included fibrous scars that replaced muscle bundles (Figure 3a). There were sparse perivascular chronic inflammatory infiltrates (results not shown). A section of mitral valve showed fibrosis of a leaflet (results not shown). 

The liver exhibited marked portal lymphocytic inflammation with the formation of lymphoid follicles (Figure 3). There was some extension of the inflammation to the periportal regions, but the inflammatory clusters were predominantly localized to portal zones. There was no necrosis and no significant fibrosis.

### 3.3. Inflammatory Infiltrates

Immunostains of the CNS, liver, kidney, and heart with CD3 and CD20 antibodies revealed that the lymphocytes were predominantly T cells (CNS Figure 4a, Kidney, Figure 4b). Only rare B cells were present (results not shown).

### 3.4. IHC/FISH/Confocal Microscopy Results

We evaluated four different autopsy tissues (heart, kidney, liver, and brain) for the potential presence of the *Borrelia* antigen using *B. burgdorferi*-specific IHC, FISH, PCR, WGS, and metagenomics methods. To investigate the potential presence of *Borrelia* biofilms in those autopsy tissues, we also utilized a biofilm-specific marker, alginate, in double immunofluorescence staining, as described previously [50]. The obtained images were analyzed by fluorescence and confocal microscopy.

In the first set of IHC experiments, 50 sections from each of the four organs (brain, heart, kidney, and liver) were immunostained with a monoclonal antibody specific for *B. burgdorferi* sensu stricto. We found evidence for *B. burgdorferi*-positive spirochetal clusters and aggregates in all organs. (Figure 5, Panels A, E, I, and M). Furthermore, using an anti-alginate antibody, we found evidence of large biofilm-like aggregates (Figure 5, Panels A, E, I, and K; green arrows) in all organs. To determine the specificity of the antibodies, nonspecific IgG was used as a negative control in all IHC experiments (Figure 5, Panels C, G, K, and O). The morphology of the tissues was visualized using differential interference contrast microscopy (DIC) (Figure 5, Panels D, H, L, and P).

To quantify our IHC findings, we next examined additional consecutive sections from each of the four organs for the number and size of *B. burgdorferi* biofilms (Table 1). A total of 250 brain tissue sections were immunostained as described above; the results showed 0–4 biofilms per slide ranging from 20–150 µm in size. In heart sections (155), we found that each section contained 0–6 biofilms, with sizes in the range of 20–100 µm. Kidney tissue sections (165) contained 0–4 biofilms/section in the range of 20–200 µm in size. We found the largest and greatest number of biofilms in the 180 liver sections with 0–7 biofilms/section in the range of 20–300 µm in size. Statistical analyses of the number of biofilm structures in the different organs, however, showed no significant differences. (*p*-values > 0.05) 

Appendix A provide more examples for the *B. burgdorferi*/alginate-positive biofilm structures in the brain, heart, kidney, and liver autopsy tissues, respectively. In these experiments, we included additional controls to ensure the specificity of our IHC procedures. For example, we performed alginate IHC on consecutive slides to demonstrate independent *Borrelia* and alginate staining on the same biofilm. Furthermore, in addition to the nonspecific IgG negative control, we also included autopsy tissues of the brain, heart, kidney, and liver from a similarly aged patient with no pathological changes in any of these organs. Those negative control autopsy tissue sections were immunostained for *B. burgdorferi*/alginate following the same IHC procedure and showed no positive staining for either *B. burgdorferi* or alginate antigens (Appendix A, Panels I, J). Furthermore, we purchased commercially available healthy tissue slides for all four organs (20 slides/each) and immunostained them for *B. burgdorferi* and alginate antigens and found no evidence of any positive staining for *B. burgdorferi* and alginate antigens in those tissue sections (data not shown). As an independent verification for the IHC experiments, brain sections from the case were sent to the Innsbruck Medical Center (Innsbruck, Austria) and IHC was performed as described previously [50]. Appendix A shows a representative image of IHC staining of brain tissue for *B. burgdorferi* (red staining with red arrows).

The spatial distribution of the alginate staining in a *Borrelia* spirochete and aggregate positive heart tissue was also investigated. Figure 6 shows that *Borrelia*-positive spirochete clusters (green staining, green arrow) do not stain for alginate, while the *Borrelia*-positive aggregate had significant amounts of alginate on the surface (blue staining, blue arrow). To prove further that the *B. burgdorferi* and biofilm marker alginate antigens were colocalized, we performed confocal microscopy analyses on IHC-positive tissue sections. Figure 7 provides a representative confocal microcopy image of a *B. burgdorferi*- and alginate-positive aggregate in liver autopsy tissue. The spatial distribution of the obtained fluorescent staining pattern confirmed the colocalization of *B. burgdorferi* and alginate antigens and showed that the alginate is on the surface on a well-formed *B. burgdorferi*-positive aggregate. 

We then looked for the presence of *B. burgdorferi* DNA associated with the biofilms in all four organs using a previously validated and published FISH procedure combined with alginate IHC [43]. FISH results showed the aggregates are positive for *B. burgdorferi*-specific 16S rDNA in all four tissues (Figure 8, Panels A, G, M, and S; green arrow). The *B. burgdorferi* DNA positive aggregates, but not the spirochetes (Figure 8, Panel G; small green arrowhead), also immunostained with the alginate antibody (Figure 8, Panels B, H, N, and T; blue arrow). DIC microscopy was used to visualize the morphology of the biofilm structure and the surrounding tissue (Figure 8, Panels C, I, O, and U). For negative controls of all FISH experiments, competing oligonucleotide probes (Figure 8, Panels D, J, P, and V), DNase I-treated samples (Figure 8, Panels E, K, Q, and Y) and a random DNA probe (Figure 8, Panels F, L, R, and Z), were used to test the specificity of the 16S rDNA probes on sequential tissue slides. None of the negative control experimental conditions resulted in any positive staining on any of the tissues studied.

We also conducted a quantitative analysis of FISH-IHC experimental data on a total of 210 brain, 130 heart, 145 kidney, and 150 liver consecutive tissue sections for the size and frequency of *B. burgdorferi* biofilms. The obtained results showed some but not statistically different variations, as we found in the IHC analyses: brain tissues contained 0–3 biofilms per slide, with a biofilm size of 20–150 μm, while heart and kidney tissues containing 0–4 biofilms per slide had a biofilm size in the range of 20–100 μm. Liver tissues contained 0–6 biofilms per slide with a biofilm size in the range of 20–300 μm (Table 2).

### 3.5. PCR/Whole-Genome Sequencing/Metagenomics Data

The next set of experiments was designed to provide further evidence for *Borrelia* DNA in the autopsy tissues. Genomic DNA purified from all four organs was used for a standard *B. burgdorferi*-specific 16S rDNA PCR analyses, as described previously [43], and the amplified DNAs were sent for direct sequencing. 

PCR results for all four organs (brain, heart, kidney, and liver) resulted in *B. burgdorferi*-positive DNA. Basic Local Alignment Search Tool (BLAST, NBCI) analyses confirmed a 99–100% identity to *B. burgdorferi* sensu strictro strains (data not shown). Because of the known problem with false PCR positivity in clinical samples [51], we also sent out the DNA samples for independent verification. Whole-genome sequencing (WGS), combined with metagenomic analyses, was performed as described in Materials and Methods at PerkinElmer DNA Sequencing and Analyzing Services (Branford, CT, USA) on genomic DNA extracted from all four tissues. Because of the high degradation of the 16+-year-old samples, only liver tissues produced the quality of DNA suitable for WGS. As a negative control, DNA samples were also obtained from normal human liver tissues (Columbia University Pathology, New York, NY, USA) and sent in parallel for the same WGS-metagenomics analyses. The Illumina WGS method produced over 186 million DNA sequencing reads (100 base long/each), which contained both human and non-human sequences. Reads were aligned to human reference GRCh37 with the Burrows‒Wheeler Aligner (BWA), V 0.6.2. (Bio-bwa.sourceforge.net, http://bio-bwa.sourceforge.net). The resulting alignment files (bam) were filtered using Samtools (Htslib.org, http://www.htslib.org) to remove reads that mapped to the human reference. The remaining reads were then filtered to remove low-quality reads using Trimmomatic (Usadellab.org, http://www.usadellab.org/cms/?page=trimmomatic). The remaining high-quality reads were then mapped, using BWA, against *B. burgdorferi* reference genome sequences, which resulted in 517 different reads. They were then further analyzed by aligning them to reference sequences for *Borrelia burgdorferi* sensu lato and sensu stricto strains using the basic local alignment search tool (BLAST) from the National Center for Biotechnology Information website (Ncbi.nlm.nih.gov, https://www.ncbi.nlm.nih.gov). Those alignments resulted in 14 sequences that mapped for different *B. burgdorferi* sensu stricto strains with >99% identity and >99% coverage. The following *Borrelia* strains matched the sequences with the same identity and coverage.

*B. burgdorferi* 382, N40, B331, JD1, HB19, Pali, and PAbe strains demonstrated the presence of *B. burgdorferi* sensu stricto and further confirmed our previous PCR data (Table 3).

### 3.6. Infiltrating CD3+ T Lymphocytes

As described above, positive immunostaining for CD3 lymphocytes was found in all four organs in the initial histopathology analyses. To follow up on this observation, we asked whether the presence of *B. burgdorferi* biofilm correlated spatially to the distribution of CD3+ lymphocytes, indicating potential local tissue inflammation next to *B. burgdorferi* biofilms. Therefore, in the next set of experiments, we analyzed *B. burgdorferi*/alginate-positive tissue sections for infiltrating CD3+ T lymphocytes from the four organs. First, we identified sections with positive immunostaining for *Borrelia* (Figure 9, Panels A, E, I, and M; green staining) and alginate (Figure 9, Panels B, F, J, and N; red staining) using IHC procedures as described above to depict biofilm structures. In good agreement with the previous IHC experiments, a nonspecific IgG isotype control was used as a negative control (Figure 9, Panels C, G, K, and O). After finding a biofilm, the adjacent slides were stained with a CD3 antibody. While heart tissues did not reveal any CD3+ cells (Figure 9, Panel H), brain, kidney, and liver tissues demonstrated aggregates of CD3+ lymphocytes surrounding the *B. burgdorferi* biofilms (Figure 9, Panels D, L, and P), but not the adjacent tissues. Interestingly, the size of the biofilm did not correlate with the severity of lymphocytic inflammation, as a very small biofilm aggregate in kidney tissue was associated with a marked lymphocytic response (Figure 9, Panel L), while a larger biofilm structure in the liver resulted in a more localized CD3 staining pattern (Figure 9, Panel P).

## 4. Discussion

### 4.1. Long-Term Persistence of *Borrelia* Antigens and DNA

In this study, we asked if there was evidence of *B. burgdorferi* in a Lyme disease patient who received extensive antibiotic treatment during the course of her 16-year-long illness. Despite this long treatment protocol, the patient experienced recurrent symptoms of Lyme disease, including neurologic deterioration, joint pain, and severe worsening of pulmonary function, and eventually died [11,47,48,49]. Examination of her liver, heart, kidney, and brain tissues revealed significant pathological changes and also provided several lines of evidence for the presence of *Borrelia* antigen/DNA in all of the tissues studied. To prove the presence of *B. burgdorferi* DNA, in addition to FISH and standard PCR, we employed an independent verification approach using WGS/metagenomics, which revealed significant amounts of *B. burgdorferi*-specific DNA sequences in liver tissues. Furthermore, IHC, FISH, and confocal microscopy revealed not only spirochetes, but also *Borrelia* aggregates, in the liver, heart, kidney, and brain.

### 4.2. *Borrelia* Antigens and DNA Are Associated with Biofilms

The presence of large numbers of *B. burgdorferi*-positive structures, as well the long-term persistence of the symptoms of Lyme disease, suggests an antibiotic-refractory chronic disease. One hypothesis for persistence is that different morphological forms of *Borrelia* may protect the bacteria from antibacterial therapy, since *B. burgdorferi* can exist in spirochetal, round body, and biofilm forms [29,30,41,42,43]. Our results support this hypothesis, showing that *Borrelia* is mainly present in the biofilm form in each organ. The *B. burgdorferi* aggregates contained the well-established biofilm marker alginate, suggesting they are true biofilms. Biofilms are aggregations of loose planktonic microorganisms that attach to biotic and abiotic surfaces and provide protection to individual cells against unfavorable environmental conditions such as pH, temperature, free radicals, antibiotics, and antimicrobials, as well as high concentrations of oxygen [33,34]. These resistant aggregations grow through continuous spreading by detaching and seeding onto surrounding surfaces, causing chronic infections [35,36,37]. In the human body, biofilm formations by pathogenic species are one of the main reasons for developing chronic diseases [37]. Biofilms are ubiquitous in normal and pathogenic human processes and they provide homeostasis in various hostile environments [33]. They are held together and protected by a barrier known as extracellular polymeric substance (EPS), consisting of proteins, nucleic acids, extracellular DNA, polysaccharides including alginate, various lectins, and glucosamines [34,41,42]. The role of the EPS layer is to protect the bacteria from a hostile environment and also enhance its adhesion to solid surfaces [34].

Several chronic infections have been associated with biofilms, such as *Pseudomonas aeruginosa*, associated with cystic fibrosis, keratitis from contact lenses, urinary tract infections from *Escherichia coli*, osteomyelitis or endocarditis by *Staphylococcus aureus*, and pulmonary infections from *Streptococcus pneumoniae*. [52,53,54,55,56]. The presence of biofilms in these conditions can affect host cellular function and can lead to inflammation and tissue damage [56,57]. 

The existence of *B. burgdorferi* biofilm was proven both in vitro and in vivo in borrelial lymphocytoma skin lesions [41,42,43]. Biofilm markers such as alginate and lectins, the key components of the EPS layer in other biofilm-forming bacteria, were also found in *Borrelia* biofilms [41,42,43].

Similar to the *P. aeruginosa* biofilm, the *B. burgdorferi* biofilm also forms to protect individual spirochetes from the immune system and from antimicrobial agents. Biofilm infections are well known to be resistant to antibiotics [58]. We have previously shown that certain antibiotics, such as doxycycline, which are very effective against spirochetes, cannot eliminate the *B. burgdorferi* biofilm, which may even increase in size upon antibiotic exposure [44,45,46]. Our recent studies demonstrated that only certain antimicrobials and a combination of antibiotics are able to reduce the size of borrelial biofilms [45,46].

Another hypothesis is an extended immune response created by the continued presence of antigenic debris or immunogenic peptidoglycans shed from *B. burgdorferi* [59,60]. However, the fact that this patient was treated extensively during an illness of 16 years argues against that idea, and it is unlikely that debris could persist for that period of time. In a rat arthritis model, while streptococcal bacterial cell wall fragments were detected for several months after their systemic administration, the level of this antigen was almost undetectable by the 90-day time point [61]. 

### 4.3. *Borrelia* DNA Persists in the Long Term

Nucleic acids are excellent analytic tools for direct diagnostic tests since DNA/RNA can be quickly cleared from the human body [62,63]. In a mouse study in which heat-killed *B. burgdorferi* was injected under the skin of mice, borrelial DNA became virtually undetectable after 8 h [64]; the same study showed that no *B. burgdorferi* genomic materials can be detected in the skin, ear, ankle, or heart tissues of mice receiving killed bacteria two and four weeks after injection. Furthermore, in a ceftriaxone-treated mouse model that investigated the persistence of non-cultivable *B. burgdorferi* by monitoring the pathogen DNA level for 12 months, the *Borrelia* DNA level initially cleared after ceftriaxone treatment but resurfaced after 12 months, suggesting persistent infection [17,23]. In another non-human primate study, viable *B. burgdorferi* were recovered by xenodiagnoses and in vivo cultures from both antibiotic-treated and untreated rhesus macaques infected with *B. burgdorferi* [20,21]. Furthermore, intact spirochetes were observed in the brain and heart of *B. burgdorferi*-infected rhesus macaques 8‒9 months after antibiotic treatment [22]. Several human studies also suggested that the chronic form of Lyme disease could be caused by a persistent spirochetal infection [25,26,27,28]. Our data are consistent with these findings and provide evidence that the presence of *B. burgdorferi* DNA after antibiotic treatment also extends to humans.

### 4.4. Long-Term Infection Is Associated with Multi-Organ Pathology

We found pathology in the CNS, PNS, liver, kidney, and heart. Much of the pathology appears to be hypoxic/ischemic in nature. Thus, there was extensive fibrous scarring in the cardiac muscle and kidney. The CNS contained infarcts of acute, subacute, and chronic pathology. We found occluded blood vessels, although there was no acute vasculitis, and widespread perivascular inflammation. There was a small focus on inflammation in one nerve root. The nervous system pathology of meningo-occlusive vascular disease, parenchymal infarcts, and perivascular mononuclear inflammation is consistent with previous reports. Mononuclear, perivascular infiltrates were present in the meninges and brain and cord parenchyma of a patient with a subacute course [65], and chronic meningitis with occlusive meningovascular disease and parenchymal infarcts in a patient with a three-year course [66]. In the latter report, the authors found images consistent with spirochetes with the Warthin‒Starry stain. In one autopsy and two biopsies of patients with CNS Lyme, Oksi et al. reported perivascular inflammation and, in the autopsy, subcortical demyelination [67]. Either CSF or brain tissue, or both, was positive for *B. burgdorferi* DNA by PCR. Other reports describe clinical and radiographic evidence for vasculitis and stroke [68,69], as reviewed in Miklossy [70]. The extensive studies we have done not only show spirochetal organisms using an anti-*Borrelia* antibody, but also *B. burgdorferi* DNA and associated biofilm.

### 4.5. T-Cell Inflammation Is Associated with Biofilm

The immune response against planktonic bacteria is very well studied for many different bacteria, including *B. burgdorferi* [71], but considerably less is known about the immune response to pathogenic biofilm [72]. Our positive IHC data for the presence of infiltrating T cells next to *Borrelia* aggregates suggest a connection between *B. burgdorferi* biofilm and the host inflammatory response and raise the question of what attracted the lymphocytes to the site of the biofilm. Interestingly, it is not necessarily *B. burgdorferi* antigens, as recent studies have shown that the EPS components of biofilm such as alginate can have strong antigenicity [73]. In a pulmonary disease study, patients with *P. aeruginosa* infection had high levels of anti-alginate antibody titers in their sera, with positive connection to the severity of the disease [73]. In another study, infected subcutaneous wounds of rabbits with *P. aeruginosa* led to delayed wound healing and a massive inflammatory response [74]. Their findings of extensive amounts of extracellular polymeric substance (EPS) in those infected wounds led to the investigation of the role of EPS in the pathogenic process. In a follow-up study, they showed that wounds infected with EPS-deficient *P. aeruginosa* mutants had shorter healing times with less inflammation, while biofilm structures were still found in wounds, suggesting the importance of EPS in inflammatory processes [75]. In a study conducted to investigate how breast implants with bacterial biofilms lead to an increase in infiltrating T lymphocytes, implants were inserted into pigs and inoculated with a human pathogen, *Staphylococcus epidermis*, a biofilm-forming bacterium [76]. IHC staining of the infected implants showed high number of biofilm structures with surrounding CD3+ lymphocytes [76]. In our study, *B. burgdorferi* biofilms were adjacent to CD3+ T lymphocytes in the brain, kidney, and liver. We are in the process of further analyzing additional inflammatory markers in those tissues to better understand the connection of *Borrelia* biofilm with the host response. We are also reanalyzing our (WGS)/metagenomics sequencing reads for additional potential pathogens in her tissues to evaluate whether (an)other infection agent(s) could have contributed to her disease.

One of the obvious limitations of this study is that it investigated the potential presence of *B. burgdorferi* in several main organs of one Lyme disease patient, not multiple patients. This patient, however, is unique because she was the first reported, culture-confirmed treatment failure patient in the United States [11]. The fact that we have available autopsy tissues from major organs from this case is extremely rare, and we are not aware of the existence of multiple autopsy tissues from any other well-documented Lyme disease case to date. Furthermore, our research group consisted of the primary physician (KBL) and the two case pathologists (JEG and JL), who had detailed knowledge of this case. Our study, however, clearly shows the urgent need for established biobanks for human biopsy and autopsy tissues from Lyme disease patients. Without those research materials, we will not able to perform fundamental studies to develop a better understanding of *B. burgdorferi* infection and the progression of Lyme disease in the human body.

In summary, we provide evidence of *B. burgdorferi*-specific antigens and DNA in the liver, heart, kidney, and brain from a well-documented Lyme disease patient who received extensive antibiotic treatment during the course of her 16-year-long illness. Findings from this study suggest that *B. burgdorferi* can persist in the human body in biofilm form even after long-term antibiotic treatment. 

## 5. Materials and Methods

### 5.1. Autopsy Tissue Procurement and Processing

An autopsy was performed five days postmortem at Columbia University Medical Center. After formalin fixation, organs were sectioned and tissues embedded in paraffin. The brain was sectioned coronally, and the brain stem and cord transversely. Multiple sections were taken from the cortex and white matter of the hemispheres, basal ganglia, thalamus, brain stem, and spinal cord, embedded in paraffin. All tissue sections were stained with hematoxylin and eosin, the CNS additionally with Luxol fast blue for myelin and Bielschowsky silver stain for axons at the Columbia University Medical Center. All tissues were also immunostained with antibodies to CD3 and CD20 using Columbia University Medical Center Pathology services.

Paraffin-embedded brain, heart, kidney, and liver sections and non-infected human autopsy tissue sections (4 μm) were forwarded to the University of New Haven’s Lyme Disease Research Laboratory after receiving approval from the Institutional Review Board of the University of New Haven. The normal tissues were obtained from human brain, heart, kidney, and liver paraffin tissue sections from a patient of the same age and gender with no neurological signs or symptoms and no neuropathological changes.

### 5.2. Immunohistochemistry (IHC)

The paraffin sections of the four organs (brain, heart, kidney, and liver) were subjected to two different IHC procedures, both validated and described previously [43,47]. One of the IHC procedures was designed to add the *B. burgdorferi* and alginate antibodies to the same tissue slide, and in the other IHC procedure the alginate antibody was added to a sequential tissue section, as described previously [43]. The only difference from the published protocol was that we used two differently labeled secondary antibodies for the alginate primary antibody. It was either labeled with a fluorescent red tag (goat anti-rabbit IgG (H+L), DyLight 594 conjugated) for the IHC experiments or a fluorescent blue tag (goat anti-rabbit IgG (H+L), DyLight 405 conjugated) for the FISH and confocal microscopy analyses.

As negative controls, human brain, heart, kidney, and liver paraffin tissue sections from a patient of the same age and gender with no neurological signs or symptoms and no neuropathological changes were stained following the same procedure as above. Additional negative controls included commercially available human brain, kidney, heart, and liver tissue sections (20 each, US Biomax, Rockville, MD, USA) were also included. Non-specific isotype IgG controls (IgG1 Isotype Control, Invitrogen, Carlsbad, CA, USA MA1-10406) were used instead of the primary antibodies as IHC negative controls.

For independent confirmation, brain tissue sections were sent to Innsbruck Medical University and immunostained with a *B. burgdorferi*-specific antibody to confirm the presence of *Borrelia spp.* utilizing the Ventana-KIT (Ventana Medical Systems, Munich, Germany), as described previously [50].

Additional IHC staining for CD3+ T cell markers used paraffin sections that were sequential to sections with positive *B. burgdorferi*. The IHC procedure for CD3 included an antigen retrieval step after the deparaffinization of the tissues. Slides were immersed into a Coplin jar with a preheated mediated antigen retrieval buffer (10 mM sodium citrate, 0.05% Tween 20, pH 6.0) at 99 °C for 10 min followed by a 20-min incubation at room temperature. Slides were then placed under slowly running tap water to remove the traces of the sodium citrate buffer for 10 min at room temperature. For enzymatic secondary staining, a Vectastain ABC HRP Kit (Peroxidase, Rabbit IgG) from Vector Laboratories (Cat#: PK-4001, Vector Laboratories, Burlingame, CA, USA) was used following the manufacturer’s protocol. Briefly, the slides were washed 4× times in 1× PBS pH 7.4 for 5 min/each and incubated with diluted normal blocking goat serum for 20 min in a humidified chamber at room temperature. After the incubation period, the slides were washed again 3× with 1× PBS pH 7.4 for 5 min/each and CD3-specific primary antibody [Abcam, Cat#: ab5690) was added to the sections, followed by incubation in a humidified chamber at 4 °C overnight. The next day the slides were washed 4× with 1× PBS pH 7.4 for 5 min/each and incubated for 30 min with diluted biotinylated secondary antibody solution in a humidified chamber at room temperature. The slides were washed again 4x with 1X PBS pH 7.4 for 5 min/each and incubated with Vectastain ABC reagent for 30 min in a humidified chamber at room temperature. A DAB Peroxidase Substrate Kit (Vector Laboratories) was used to visualize the positive reaction with colorimetric stain (brown stains). The sections were counterstained with hematoxylin (BBC Chemical, Mount Vernon, WA, USA) and mounted with Permount medium (Fisher Scientific, Hampton, NH, USA). Images were taken using a Leica DM2500 microscope (Leica, Wetzlar, Germany) at 100×, 200×, and 400× magnifications.

### 5.3. Combined IHC and Fluorescent In Situ Hybridization (FISH)

Paraffin sections of the four organs (brain, heart, kidney, and liver) were subjected to a combined IHC and FISH method, which was validated and described previously [43]. Control experiments for FISH included random probe (5’-FAM-GCATAGCTCTATGACTCTATACTGGTACGTAG-3’), competing oligonucleotides for *Borrelia* (5’-CAAACGGGGAATAATTATCTCTAACTATATCC-3’), and DNase I-treated samples following the previously published protocol (43).

### 5.4. DNA Extraction/PCR

Genomic DNAs were extracted from paraffin-embedded tissues using a Qiagen QIAamp DNA formalin-fixed, paraffin-embedded (FFPE) tissue kit (Qiagen, Hilden, Germany). First, the slides were deparaffinized as described above and tissues present on the slides were scraped off using sterile blades and collected in sterile microcentrifuge tubes. Then 180 µL of ATL buffer and 20 µL of proteinase K were added to the tubes and the samples were incubated overnight at 42 °C and then at 90 °C for 1 h to deactivate the enzyme. Next 200 µL of AL buffer and an additional 200 µL of 96% ethanol were added to the samples and vortexed thoroughly. The samples were then transferred to DNeasy mini columns and centrifuged for one minute at 6000x g. The flow-through was discarded and the columns were placed into new collection tubes. To wash away any unwanted materials that might be present along with the DNA, 500 µL of AW1 and AW2 buffers were added to the columns consecutively and centrifuged at 6000x g for 1 min. The spin columns were then placed in fresh collection tubes and centrifuged at 20,000x g for 3 min to dry the columns completely. The columns were then placed in new microcentrifuge tubes and the samples were eluted twice with 50 µL of ATE buffer/each step. The DNA samples were quantified using a BioTek (Winooski, VT, USA) Microplate Spectrophotometer. The DNA samples were initially tested with standard PCR using primers designed to amplify *B. burgdorferi* 16S ribosomal DNA using published primers: F: 5’- CCTGGCTTAGAACTAACG-3’; R: 5’-CCTACAAAGCTTATTCCTCAT-3’. The PCR reaction (50 μL) contained a HotStarTaq buffer (Qiagen), 1.5 mM MgCl_2_ 25 pmoles of each primer, and 2.5 units of HotStarTaq DNA polymerase (Qiagen) in a 50 μL volume. The PCR conditions were: initial denaturation at 94 °C for 15 min, followed by 40 cycles of 94 °C/30 s, 50 °C /30 s, 72 °C/1 min, then a final extension at 72 °C /5 min. The PCR products were analyzed by standard agarose gel electrophoresis and PCR products were purified using the QIAquick PCR purification kit (Qiagen) according to the manufacturer’s instructions. Samples were eluted twice in 30 μL and the eluates from each sample were pooled and sequenced in 2× in both directions using the primers that generated the products. Sequencing reactions were performed by Eurofins/MGW/Operon (Huntsville, AL, USA).

### 5.5. Whole-Genome Sequencing/Metagenomic Analyses

DNA samples were first evaluated using an eGel and PicoGreen fluorometry at PerkinElmer DNA Sequencing and Analyzing Services to measure the quality and quantity, respectively. DNA samples were then physically sheared to the desired size (300–1000 bp) using a Covaris (Woburn, MA, USA). E220 focused ultrasonicator. Whole-genome libraries were prepared using an Illumina (San Diego, CA, USA) TruSeq kit following the manufacturer’s instructions. Whole-genome sequencing was performed using an Illumina HiSeq2500 instrument. Raw sequencing data were converted to FASTQ format using Illumina’s BCL2FASTQ v. 1.82. Reads were aligned to human reference GRCh37 with the Burrows‒Wheeler Aligner (BWA), v. 0.6.2. (http://bio-bwa.sourceforge.net). The resulting alignment files (bam) were filtered using Samtools (http://www.htslib.org) to remove reads that mapped to the human reference. The remaining reads were then filtered to remove low-quality reads using Trimmomatic (http://www.usadellab.org/cms/?page=trimmomatic). The remaining high-quality reads were then mapped, using BWA, against various bacterial reference genomes of interest. Reads that mapped to the genomes of interest were then used for BLAST searches.

### 5.6. Confocal Microscopy

The tissue sections were first immunostained for *Borrelia* and alginate, as described above, and further analyzed with a confocal scanning laser microscope (Leica DMI6000). ImageJ software (Imagej.nih.gov, https://imagej.nih.gov/ij/index.html was used to process the obtained z stacks to provide a detailed analysis of the spatial distribution of the different antigens (Plugins: Interactive 3D Surface Plot and Volume Viewer).

### 5.7. Statistical Analysis

Statistical analysis was performed using Student’s *t*-test (Microsoft Excel, Redmond, WA, USA) on the numbers of observed aggregates found in the autopsy tissues of the brain, heart, kidney, and liver. Statistical significance was determined based on *p*-values < 0.05.

## 6. Conclusions

In summary, this study provides several lines of evidence that *Borrelia* can persist in the human body not only in the spirochetal but also in the antibiotic-resistant biofilm form, even after long-term antibiotic treatment. The presence of infiltrating lymphocytes in the vicinity of *B. burgdorferi* biofilms suggests that the biofilm might trigger chronic inflammatory responses. 

## Figures and Tables

**Figure 1 antibiotics-08-00183-f001:**
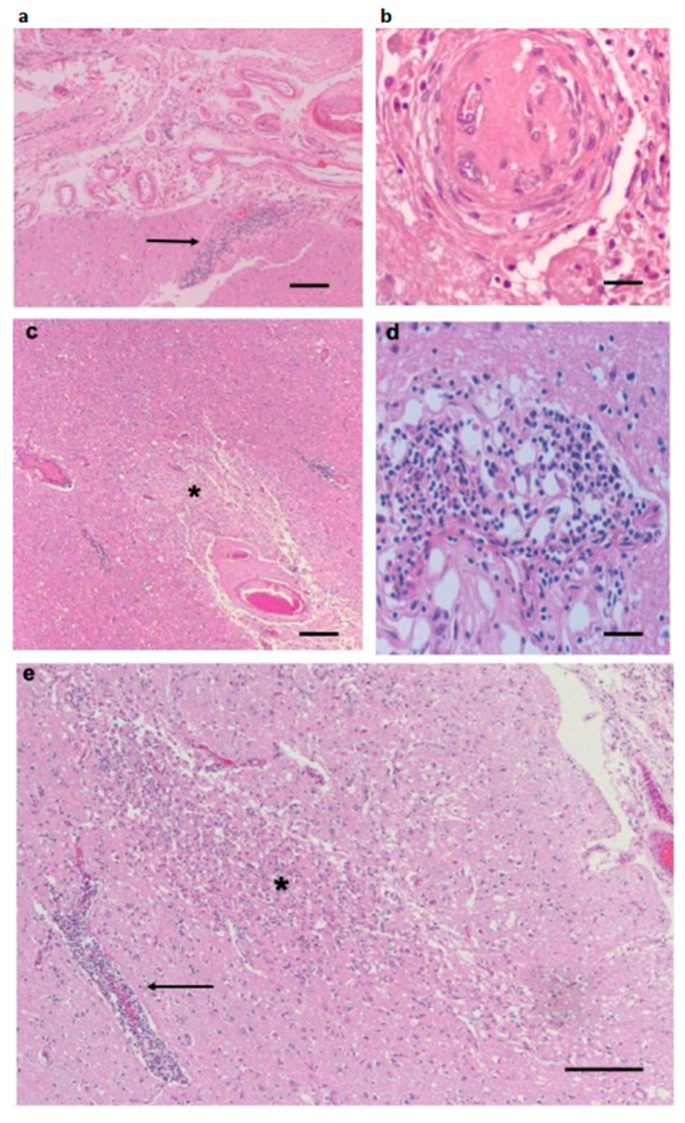
Representative pathological images of CNS. (**a**) A section through the frontal leptomeninges shows meningeal thickening and chronic inflammatory infiltrates. There is an infiltrate surrounding a penetrating vessel (arrow). (**b**) A leptomeningeal vessel that has been occluded, showing small zones of recanalization. (**c**) Deep frontal white matter contains infarcts, one illustrated here (*). (**d**) A blood vessel in the frontal deep white matter shows a mixture of lymphocytes and plasma cells. (**e**) An infarct in the frontal cortex (*) near a blood vessel with a chronic inflammatory infiltrate (arrow). All sections stained with H&E. Scale bars: (**a**) 100 μm, (**b**) 25 μm, (**c**) 100 μm, (**d**) 25 μm, and (**e**) 200 μm.

**Figure 2 antibiotics-08-00183-f002:**
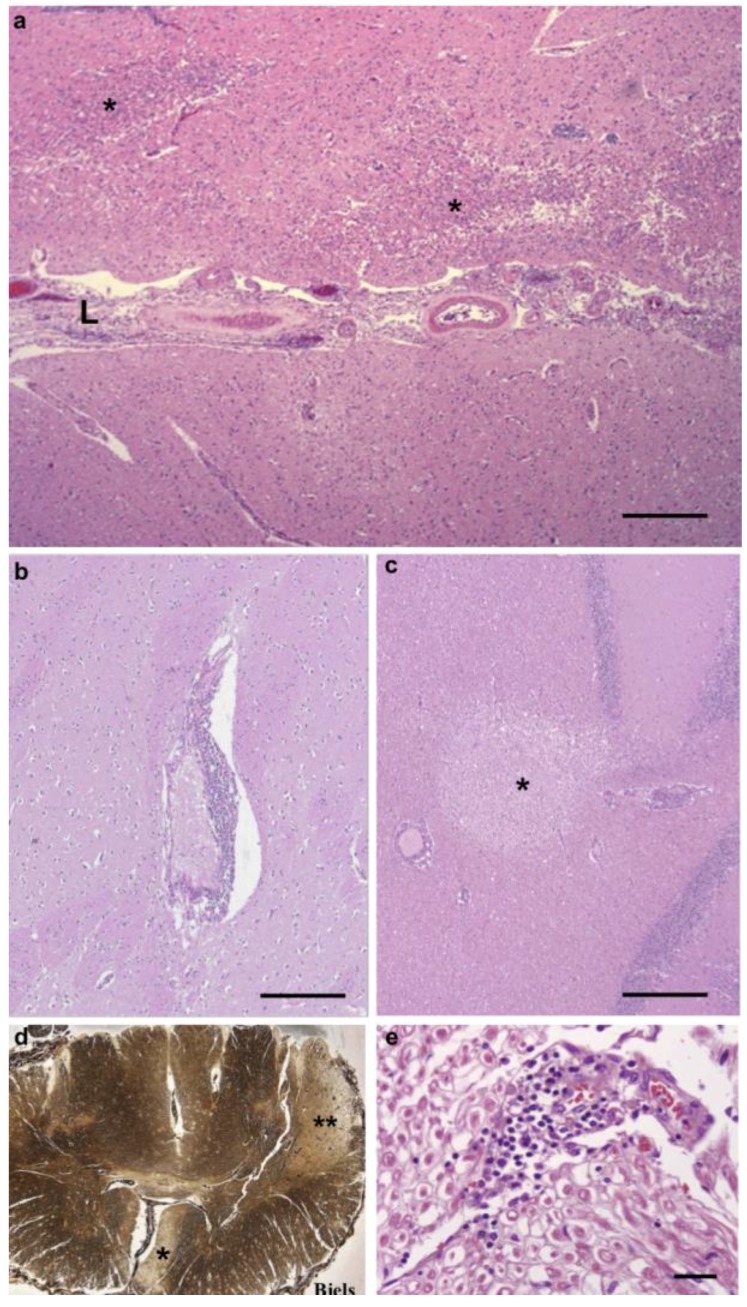
Representative pathological images of CNS and PNS. (**a**) A section of frontal cortex and leptomeninges (L) shows thickened, inflamed meninges and adjacent cortex with infarcts (*). (**b**) A large perivascular chronic inflammatory infiltrate in the putamen (arrow). (**c**) An infarct in the cerebellar white matter and cortex (*). There is a vessel with a surrounding chronic inflammatory infiltrate at the left edge of the infarct. (**d**) A Bielschowsky silver stain of an entire transverse section of the spinal cord shows severe, unilateral corticospinal tract degeneration, both lateral (**) and anterior (*) tracts (Bielshowsky silver stain). (**e**) A section of a cranial nerve shows a chronic inflammatory infiltrate associated with small vessels in the perineurium. The degree of myelination of surrounding axons appears normal. Autopsy tissues are stained with standard H&E histological stains. Scale bars: (**a‒c**) 200 μm, (**e**) 25 μm.

**Figure 3 antibiotics-08-00183-f003:**
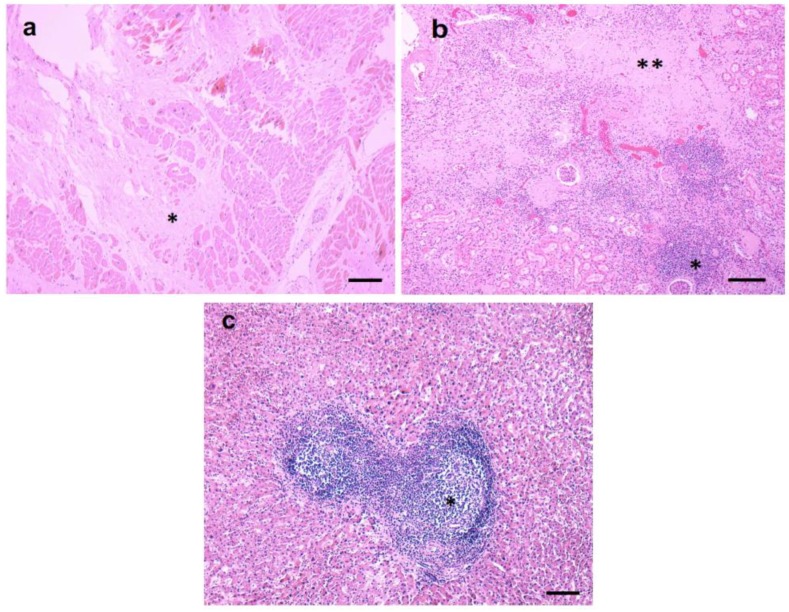
Representative pathological images of the kidney. All sections stained with standard H&E histological stains. (**a**) A section of the left ventricle with extensive fibrous scarring (*). Scale bar: 200 μm. (**b**) A section of kidney shows inflammatory infiltrates (*) scattered among glomeruli and tubules and areas of fibrous scarring (**). Scale bar: 100 μm. (**c**) A section of liver with inflammation in the portal zone (*). Scale bar: 100 μm.

**Figure 4 antibiotics-08-00183-f004:**
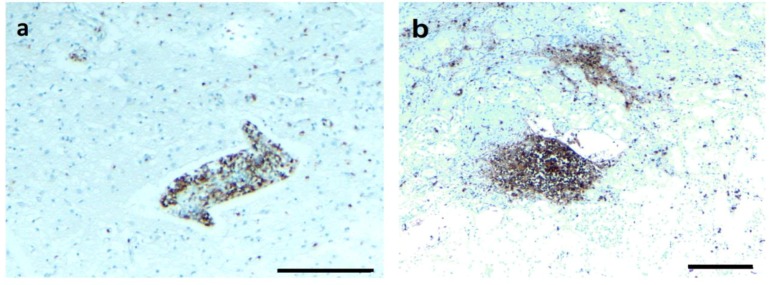
Representative images of CD3 staining in CNS and kidney tissue section. (**a**) A brain section immunostained for T cells (CD3) with a perivascular infiltrate and scattered T cells in the parenchyma. (**b**) A kidney section immunostained for T cells (CD3) shows a T cell aggregate and scattered T cells in the tissue. Scale bars: 100 μm.

**Figure 5 antibiotics-08-00183-f005:**
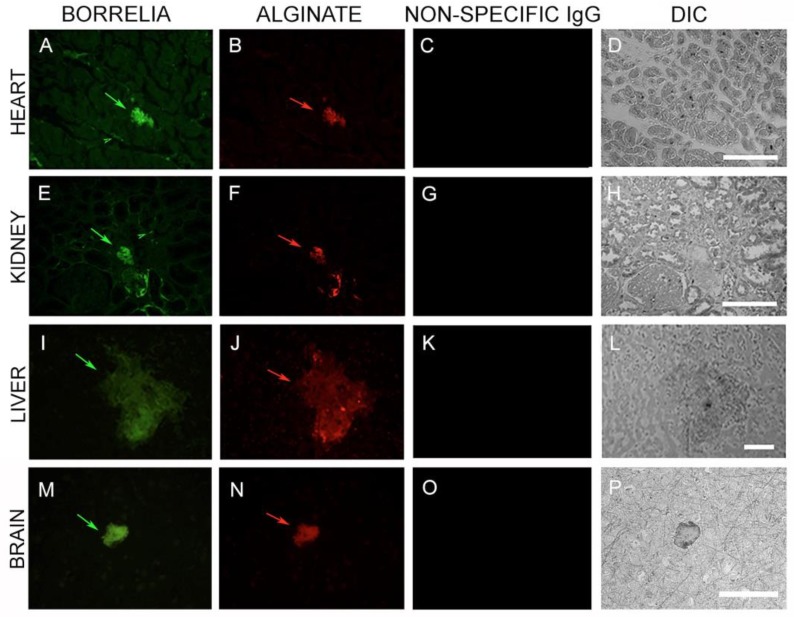
Representative images of IHC staining of autopsy tissues (heart, kidney, liver, brain) with *Borrelia*- and alginate-specific antibodies. Aggregates stained positive with *Borrelia*- (green staining: **A**,**E**,**I**,**M**) and alginate-specific antibodies (red staining: **B**,**F**,**J**,**N**) respectively. A nonspecific IgG antibody was used as an additional negative control for the primary antibodies using the sequential tissue section (**C**,**G**,**K**,**O**). Differential interference microscopy (DIC) showing the size and tissue morphology (**D**,**H**,**L**,**P**). Scale bar: 200 μm.

**Figure 6 antibiotics-08-00183-f006:**
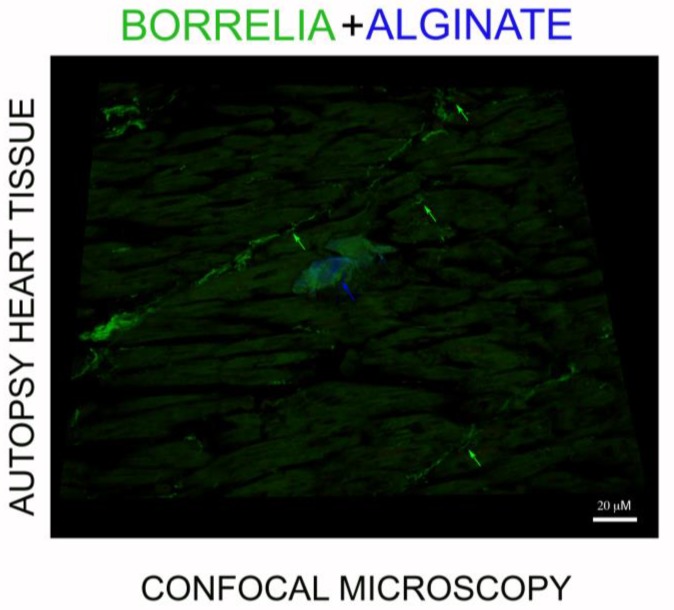
A representative three-dimensional (3D) confocal image of a *Borrelia* spirochete and *Borrelia* biofilm-positive heart autopsy tissue. Heart autopsy tissue section was immunostained with antibodies for *Borrelia* (green) and alginate (blue) and analyzed by confocal microscopy using individual z-stacks to obtain a composite 3D view (Image J) of the spatial distribution of *Borrelia* spirochetes (green arrows) and *Borrelia* aggregates with alginate on the surface (blue arrow) in the infected heart tissue. Scale bars: 20 μm.

**Figure 7 antibiotics-08-00183-f007:**
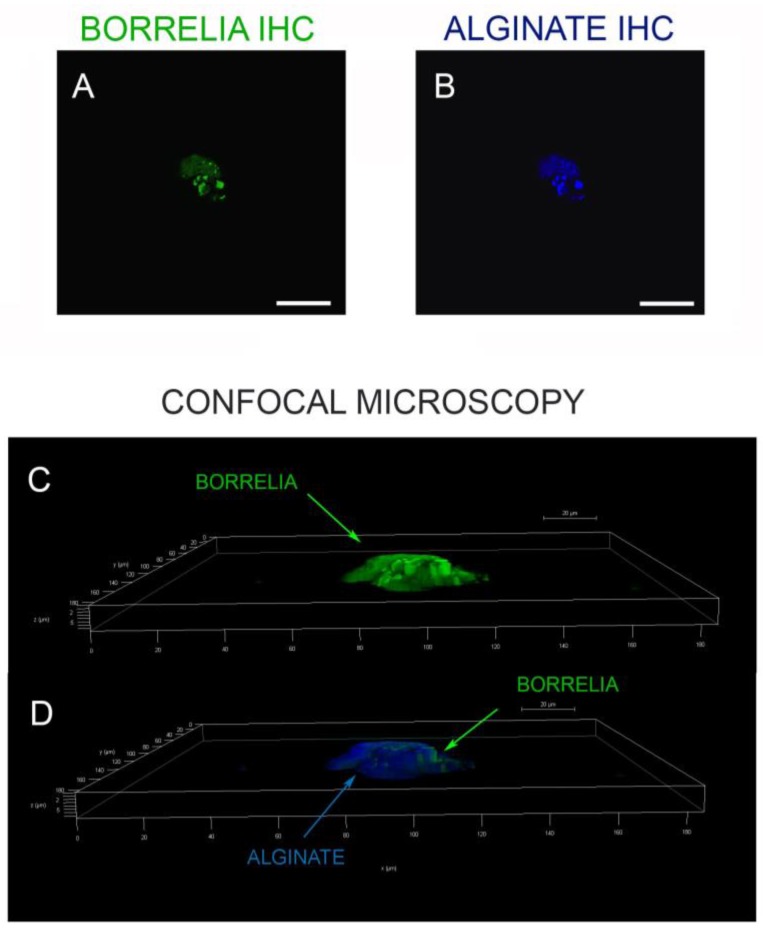
Three-dimensional (3D) analyses of a *Borrelia* biofilm in infected liver tissue. (**A**,**B**) Fluorescent microscopy images of a liver tissue section positively immunostained with antibodies against *Borrelia* (green) and alginate (blue) antigens. Scale bar: 100 µm. Confocal microscopy analyses of the same tissues section were performed using individual z-stacks to form a composite 3D image (Image J) to illustrate the spatial distribution of *Borrelia* biofilm (**C**) and *Borrelia* biofilm with surface alginate (**D**). Scale bar: 100 µm.

**Figure 8 antibiotics-08-00183-f008:**
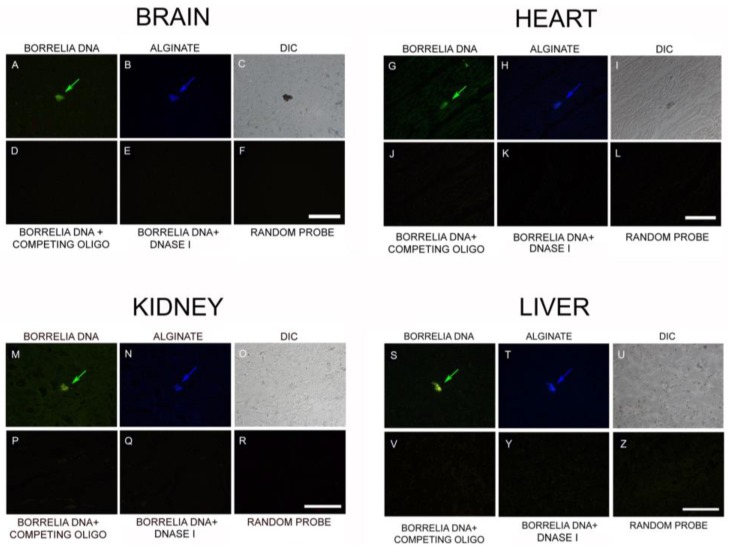
Representative images of presence of *Borrelia* biofilm DNA using fluorescent in situ hybridization (FISH) combined with alginate IHC in brain, heart, kidney, and liver autopsy tissues. The FISH results showed positive staining for *Borrelia* DNA in all four organs (**A**,**G**,**M**,**S**; green arrow) using *Borrelia*-specific 16S rDNA probes. The *Borrelia*-positive aggregates, but not the spirochetes (**G**: small green arrowhead), are also stained with alginate antibody (**B**,**H**,**N**,**T**; blue staining). Differential interference contrast (DIC) microscopy (**C**,**I**,**O**,**U**) was used to visualize the morphology of the biofilm structure and the surrounding tissue. For negative controls of all FISH experiments, competing oligonucleotide probes (**D**,**J**,**P**,**V**), DNase I-treated samples (**E**,**K**,**Q**,**Y**), and a random oligonucleotide probe (**F**,**L**,**R**,**Z**), were used to test the specificity of the 16S rDNA probes. Scale bars: 200 μm.

**Figure 9 antibiotics-08-00183-f009:**
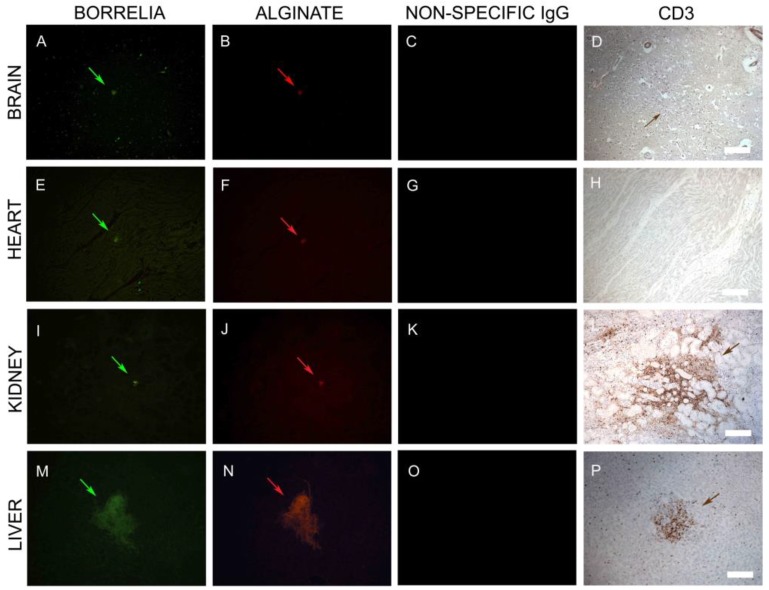
Representative IHC images of *Borrelia*, alginate, and CD3+ T lymphocytes staining in infected brain, heart, kidney, and liver autopsy tissue sections. Tissues sections that had positive staining for *Borrelia* (green staining: **A**,**E**,**I**,**M**) and alginate (red staining: **B**,**F**,**J**,**N**) were subjected to additional IHC analyses by immunostaining the sequential sections with a T cell marker, CD3-specific antibody (brown staining: **D**,**H**,**L**,**P**). Fluorescent images were taken at 100x magnification to illustrate a larger section of the tissue. CD3-positive lymphocytes surrounded these aggregates, as depicted with brown staining in the brain, kidney, and liver tissues (**D**,**L**,**P**). There was no presence of CD3-positive lymphocytes in the heart tissues (**H**). Nonspecific IgG antibody was used as a negative control for the primary antibodies (**C**,**G**,**K**,**O**). Scale bar: 100 μm.

**Table 1 antibiotics-08-00183-t001:** Quantitative analysis of the IHC experiments on *Borrelia* biofilms in brain, heart, kidney, and liver tissues. SD = standard deviation.

Organ	Number of IHC Stained Slides	Number of Biofilms per Slide ± SD	Size of the Biofilm (µm)
Brain	250	0–4 ± 1.2	20–150
Heart	155	0–6 ± 1.5	20–100
Kidney	165	0–4 ± 1.1	20–200
Liver	180	0–7 ± 1.6	20–300

**Table 2 antibiotics-08-00183-t002:** Quantitative analysis of the FISH experiments on *Borrelia* biofilms in brain, heart, kidney, and liver tissues.

Organ	Number of IHC Stained Slides	Number of Biofilms per Slide ± SD	Size of the Biofilm (µm)
Brain	210	0–3 ± 1.1	20–150
Heart	130	0–4 ± 1.2	20–100
Kidney	145	0–4 ± 1.1	20–100
Liver	150	0–6 ± 1.5	20–300

**Table 3 antibiotics-08-00183-t003:** The actual sequences, the gene/genomic regions, and the percentage of coverage/identity with E values. There were an additional 20 sequencing reads out of 517 reads which were matches for other *Borrelia* strains with >90% identity and coverage, but they also had similar identities to other bacterial species (data not shown). For the control liver, eight sequences came through the metagenomics analyses pipeline for *B. burgdorferi*, but they did not match to any *Borrelia* species.

Whole Genome Sequencing Reads For Borrelia Burgdorferi Sensu Stricto Strains	Gene/Region	Coverage	Identity	E value
AGCTTTGCTATCTCAAATGTCAAAGACTCTATCTCTTCTTGAGAAAGATACTTAAACACTTTAGAAGAGATTTCAGAACCTATTGAAACCAACAAAATAG	Flagella switch protein (fliG)	100%	100%	3e−42
AAAACTATTAAAATTACCCTTAACAATTGCAATGTAAACTTTATTTGTTCTTTTATCTTTAAACTGCTGAGCTAAAAATCTTAAGGTGCTAATGTTTTTT	Helicase protein(Yfi)	100%	99%	1e−40
AAGGTCTTATGCCAATAAAAATCCAATCACAGAATACAAAGAAGAGGGATTTTCAATATTTAGCGAGCTTATTAAAGATATTAAAGTTTCTACCATAAGG	SecA protein	100%	100%	3e−42
AAGAAAAGATTTTCCTATTTTAAATAAAAAATTTGACAATAAGTATATAATTTACTTTGATAATGCAGCAACCTCTCAAAAGCCCAAAAACGTAATTTAT	M11p aminotransferase (nifS)	100%	100%	3e−35
ATTACAGCGTTACTGTTTTAATGAAGCAATTGCCATACTATCAAAACCAATTAGCATTTATCATGAAAGATGTGCTTAGTCGATATAAAGTTGATAGTTC	Left subtelomeric chromosomal region	100%	100%	3e−42
TCATTTCAAAAACATGTATTTCTGAAAGCAAAAAATACAACAGCAAAAAAACTACTACCAAACTGCTTGTAAATCCAATAATTTCATTATAAGCTCTTGT	Left subtelomeric chromosomal region	100%	100%	3e−42
TTGAATATTTTGAAATAACTTATGAGGCTTATGCTCCTTATGGAGTGGCTCTAATGATTAAATGCTTAACGGATAATAAAAACAGAACCTCTAGCGATGT	Intergenic region	100%	100%	3e−42
AAGAAGAATTAGAAGTTTGCGAGCTAAATGGAAAAGATTGGACATTAAAATTTAAAAAACCGCTAAAAGCATATAAATTCTTAAAATCCGTAGGAAG	Intergenic region	99%	100%	1e−40
TTACTAAAACTTCAGAAGAGCCCCTAATGCTTGTTTTAATGATAGGCATTATTTCTTTGGCCTGTTGATAGTCTATGTTTGTGTATGTATTGTTATTCAT	Intergenic region	100%	99%	1e−40
AATCTTAAAATTAAAAGATAACGACAAATTTAAATTTGGTATTCTTGGAGAAAAAAACATTTACCACTGCATTTACAAAAAAGATAAAAAACTATTTTTC	Intergenic region	100%	100%	3e−42
TAAGTTATAATTGAGGAATAATAGCAAATATTTTAACTTTTTGGTATAAATTACTACTAGATTTATATGTTAAGTTTTGCGAGGTATTTAAATGGCAGTA	Intergenic region	100%	99%	1e−40
CAAGAGTTAGTATTGGCCTTAAAAAACGATAAAGTTGATTATATATATGGTGATTGCAAGACTTTACATTATATTGCAAATAACTTTTTAAGTGA	Intergenic region	100%	100%	1e−39
CTTGAGGGATTTAAAGAAGTTAAGCCTGTAGTATTCTCTTCAGTTTATCCGTTGATGCTAATCAATATGATGATCTTTTAAGGGCAATGGATAGATTAA	Intergenic region	100%	99%	1e−40
TTTATACTAATAAACTTTCAATTTCTTTTGTGAAGATATTGAAAGAAATCCATGTCTGTTGAGAAAATTTTTCTTTTATCTTTTAATACTGCTTTATAGC	Intergenic region	100%	100%	3e−42

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
