# Peer review of "The Long-Term Persistence of *Borrelia burgdorferi* Antigens and DNA in the Tissues of a Patient with Lyme Disease"

_antibiotics, 2019, doi:10.3390/antibiotics8040183_

Round 1

Reviewer 1 Report

This is a superb presentation showing the persistence of Borrelia in a patient with a downward spiralling course of Borreliosis. Because of the development of biofilms, the organism survived and severely damaged tissues and ultimately led to the patient's demise.

The methods and procedures were superbly carried out; particularly the 3D confocal assays and the WGA analyses. These will likely become "state of the art" protocols. Even the H+E images will add to a field that is notably barren. 

Also important is inserting some rigorous science to challenge the "post treatment Lyme disease syndrome". This work shows that to be microbial. 

This effort also shows the absolute need for inclusion of a biofilm dispersing agent in future treatment protocols.

Author Response

Thank you for your positive review, we are very happy that you have found the study important and helpful for future treatment protocols of Lyme disease.

Reviewer 2 Report

Summary: This research describes the detection of Borrelia burgdorferi in tissue samples taken from a patient who was extensively treated for Lyme disease, remained symptomatic during treatment and died after antibiotics were withheld as a result of loss of insurance reimbursement for continuing antibiotic treatment. Tissues tested were brain, heart, kidney and liver, using direct detection of Borrelia spirochetes and biofilms by histology and immunohistology with confocal microscopy. In addition, B. burgdorferi DNA was detected by FISH, by PCR, and confirmed by metagenomic sequencing. Inflammatory infiltrates of CD3+ T-cells were also noted in all tissues. This study documents that B. burgdorferi may remain present in tissues during antibiotic treatment and implicates persistent infection as a driver of inflammation in infected tissues.

Strengths: This study provides overwhelming evidence that symptoms of Lyme disease experienced by a significant number of Lyme disease patients after antibiotic treatment, particularly if the initial antibiotic treatment is not initiated early in the infection (as it was not in this case), are inflammatory in nature and are the result of colonization of tissues by Borrelia. PCR results were independently and externally confirmed by metagenomics sequencing. The diversity of laboratory methods and appropriate inclusion of controls used provide sufficient data to assure that the results could not be due to laboratory error or contamination.

Specific Comments:

Line 291: Change to “… did not match to any Borrelia species.”

Line 345-6: Cystic fibrosis isn’t “caused” by Pseudomonas, it’s a genetic disorder complicated by infection with Pseudomonas.

Line 354: italicize “Pseudomonas”

Line 489: Line is formatted awkwardly

Author Response

Thank you for your positive review, we are very happy that you have found that this study had a potential to advance our understanding of the antibiotic resistance of Borrelia burgdorferi. We have agreed on all suggested changes and they can be found in the revised manuscript marked with red color.

Line 291: Change to “… did not match to any Borrelia species.” It was changed and can be find in line 295 of the revised manuscript.

Line 345-6: Cystic fibrosis isn’t “caused” by Pseudomonas, it’s a genetic disorder complicated by infection with Pseudomonas. The sentence was corrected and can be found in lines 348/349 of the revised manuscript.

Line 354: italicize “Pseudomonas” It was corrected and found in line 360 of the revised manuscript.

Line 489: Line is formatted awkwardly – corrected line 494 of the revised manuscript.

Reviewer 3 Report

I’m very interested in the importance of this manuscript, but I strongly believe that there are necessary several changes before the acceptance of this manuscript. See bellow:

Abstract: Please, provide some information before the objectives of the study. Moreover, it would be necessary to summarize the objectives. Therefore, I would also recommend that authors should follow this structures for an abstract: introduction, aims, methods, results and discussion. Format issues: the size of the letters is different from first and second paragraph (p. 2). There are some information in the introduction that I think would be appropriate for participants description. In fact, this information starts after the second paragraph of the introduction. It would be nice to follow a similar order for the exposure of each patient. There is also necessary to provide additional information regarding demographic characteristics of each patient. There is completely necessary to restructure the results in order to follow a logic order. In this sense, it would necessary to introduce a small presentation of the results instead of enumerate. Moreover, I think that it is necessary links each results with each patient. This will affects the presentation of the discussion.

Author Response

Thank you for acknowledging the potential significance of this study and thank you for your very valuable suggestions. We have agreed on all suggested changes which can be found in the revised manuscript marked with red color.

Abstract: “Please, provide some information before the objectives of the study. Moreover, it would be necessary to summarize the objectives. Therefore, I would also recommend that authors should follow this structures for an abstract: introduction, aims, methods, results and discussion”

We agree that the abstract can use a more structured format, therefore we updated it by following your suggestion.

Format issues: the size of the letters is different from first and second paragraph (p. 2).

We agree, it was an inconsistency in the format and it was corrected in the revised manuscript.

“There are some information in the introduction that I think would be appropriate for participants description. In fact, this information starts after the second paragraph of the introduction. It would be nice to follow a similar order for the exposure of each patient. There is also necessary to provide additional information regarding demographic characteristics of each patient. There is completely necessary to restructure the results in order to follow a logic order. In this sense, it would necessary to introduce a small presentation of the results instead of enumerate. Moreover, I think that it is necessary links each results with each patient. This will affects the presentation of the discussion”

This study investigated the potential presence of Borrelia burgdorferi in several main organs of one Lyme disease patient not multiple patients. This patient however is very unique because it is the first reported culture-confirmed treatment failure patient in United States. The fact we have available autopsy tissues from major organs from this case is extremely rare and we are not aware of the existence of multiple autopsy tissues from any other well documented Lyme disease case to date. Furthermore, our research group consisted of the primary physician and case pathologists who were able guide us with detailed knowledge of this case.

That is the main reason we presented the results in a structured format to provide objective view of pathological and cellular and molecular biology findings.

The manuscript needs to be checked for the correct English and grammar.

Thank you for pointing out that we need to check the manuscript for the correct English and grammar. We checked the entire manuscript for grammar, spelling, and punctuation issues and corrected them (marked with red color in the entire manuscript).

Round 2

Reviewer 3 Report

Abstract:

1) Authors need to provide an introduction to their abstract. In fact, they repeated "objectives" and "aims". Moreover, I would recommend that the size of each part could be equal.

2) Additional information regarding participant (i.e. demographics, disease...).

Introduction:

3) It would be necessary to provide additional information of Lyme disease, instead of describe the case in the second paragraph. It seems odd and redundant to repeat the same information.

Methods:

4) I would like to see that authors provide more information regarding cognitive status of patient (i.e. changes over time?...).

Discussion:

5) Differences in letter size ("during the course of her 16 year-long illness.").

6) It is completely necessary to limitations and future directions of your study.

Author Response

Thank you again for your very valuable suggestions. We have agreed on all suggested changes which can be found in the revised manuscript marked with red color.

1) Authors need to provide an introduction to their abstract. In fact, they repeated "objectives" and "aims". Moreover, I would recommend that the size of each part could be equal. We agree and we added an introductory sentence to the abstract and reformatted the abstract structure accordingly (marked with red color).

2) Additional information regarding participant (i.e. demographics, disease...). We agree it is a very important information and we added this information directly to the abstract (marked with red color).

Introduction:

3) It would be necessary to provide additional information of Lyme disease, instead of describe the case in the second paragraph. It seems odd and redundant to repeat the same information. We agree, it is redundant, therefore we restructured the entire introduction and we even removed information which can be found in the clinical history part right below the introduction.

Methods:

4) I would like to see that authors provide more information regarding cognitive status of patient (i.e. changes over time?...). We agree it is a very important information. We added a whole paragraph into the clinical history section. Lines 126-135, marked with red color.

Discussion:

5) Differences in letter size ("during the course of her 16 year-long illness."). We checked on it and it was different font style (Times, not Palatino Linotype) that is why it looked different. Thank you for catching this error. We fixed it.

6) It is completely necessary to limitations and future directions of your study. We agree that it will strengthen the discussion of the manuscript and we added a whole paragraph at the end of discussion about the limitations of the study and future directions. Lines 424-437, marked with red color.

We are resubmitting this manuscript and hope that the changes, as recommended,  now render it suitable for publication in “Antibiotics”.

Round 3

Reviewer 3 Report

It has been improved the quality of the manuscript. Hence, I think it is suitable for publication.